



# Synoptic conditions and atmospheric moisture pathways associated to virga and precipitation over coastal Adélie Land in Antarctica

Nicolas Jullien[1], Étienne Vignon[1], Michael Sprenger[2], Franziska Aemisegger[2], and Alexis Berne[1]

[1]Environmental Remote Sensing Laboratory (LTE), École Polytechnique Fédérale de Lausanne (EPFL), Switzerland
[2]Institute for Atmospheric and Climate Science, ETH Zürich, Zürich, Switzerland

**Correspondence:** Alexis Berne (alexis.berne@epfl.ch)

**Abstract.**

Precipitation falling over the coastal regions of Antarctica often experiences low-level sublimation within the dry katabatic layer. The amount of water that reaches the ground surface is thereby considerably reduced. This paper investigates the synoptic conditions and the atmospheric transport pathways of moisture that lead to either virga - when precipitation is completely

sublimated - or actual surface precipitation events over coastal Adélie Land, East Antarctica. For this purpose, the study combines ground-based lidar and radar measurements at Dumont d'Urville station (DDU), Lagrangian back-trajectories, Eulerian diagnostics of extratropical cyclones and fronts as well as moisture source estimations. It is found that precipitating systems at DDU are associated with warm fronts of cyclones that are located to the west of Adélie Land. Virga - corresponding to 36% of the hours with precipitation above DDU - and surface precipitation cases are associated with the same precipitating system but

they correspond to different phases of the event. Virga cases more often precede surface precipitation. They sometimes follow surface precipitation in the warm sector of the cyclone's frontal system, when the associated cyclone has moved to the east of Adélie Land and the precipitation intensity has weakened. On their way to DDU, the air parcels that ultimately precipitate above the station experience a large-scale lifting across the warm front. The lifting generally occurs earlier in time and farther from the station for virga than for precipitation. It is further shown that the water contained in the snow falling above DDU

during pre-precipitation virga has an oceanic origin farther away (about 30° more to the west) from Adélie Land than the one contained in the snow that precipitates down to the ground surface.

## 1 Introduction

Precipitation is the main water input to the mass balance of the Antarctic ice sheet (e.g., King and Turner, 1997). The atmospheric synoptic circulation over the Austral Ocean drives the advection of moisture from the mid-latitudes towards Antarctica

(Bromwich et al., 1995; Grieger et al., 2018), thereby controlling the amount and location of precipitation falling above the continent (Souverijns et al., 2018). Extratropical cyclones are responsible for a very important part of this moisture transport (Simmonds and Keay, 2000; Simmonds et al., 2003; Uotila et al., 2013). In cyclones, the poleward transport of moisture and the precipitation production mostly occurs in the warm and moist air stream that rises along the cold front within the cyclone's warm sector (the so-called 'warm conveyor belt', Eckhardt et al., 2004; Madonna et al., 2014). Sinclair and Dacre (2019)





underline that the quantity of water extratropical cyclones bring to Antarctica mostly depends on the latitude of cyclogenesis and on the poleward propagation speed: the closer to the equator cyclones form and the faster they transit to Antarctica, the more they carry water onto the ice sheet. During atmospheric blocking conditions (Naithani et al., 2002; Schlosser et al., 2010; Hirasawa et al., 2013), the advection of moisture along the eastern flank of cyclones can be particularly long and intense and

can sometimes be associated with so-called 'atmospheric rivers' (Gorodetskaya et al., 2014; Wille et al., 2019). Gorodetskaya et al. (2014) show that atmospheric rivers are responsible for anomalously high snow accumulation events over Dronning Maud land, East Antarctica, impacting substantially the surface mass balance in this region. More generally, Turner et al. (2019) reveal that a large proportion of the total precipitation amount over the Antarctic ice sheet is brought by extreme precipitation events.

While the synoptic dynamics governs the advection of moisture over the ice sheet, the mesoscale dynamics exerts a significant control on the spatial distribution of precipitation as well as on the amount of precipitation that reaches the ground surface. In particular, Grazioli et al. (2017b) show that the relatively dry low-level katabatic winds blowing from the high Plateau towards the ice sheet edges (Parish and Bromwich, 2007) sublimate a substantial part of precipitation falling from overlying clouds. Such result were confirmed and further investigated in Vignon et al. (2019c) by inspecting the vertical profiles of rel-

ative humidity from radiosoundings over the coastal margins of East Antarctica. Atmospheric reanalysis and regional models suggest that about 17 % of precipitation sublimates at the continental scale (Grazioli et al., 2017b; Agosta et al., 2019), this percentage reaches about 35 % over the coastal regions.

Adélie Land, East Antarctica, experiences both fierce katabatic winds (Wendler et al., 1997) and strong precipitation sublimation (Vignon et al., 2019c, a). Inspecting vertical profiles of reflectivity from a precipitation radar, Durán-Alarcón et al.

(2019) further reveal that 36 % of the time when precipitation is detected in the column above Dumont d'Urville (DDU) station actually corresponds to virga cases, i.e. cases during which no precipitation reaches the ground surface. Such a percentage is comparable to the one estimated for precipitation falling over deserts in arid tropical regions (Wang et al., 2018).

Overall, understanding when, where and how precipitation is sublimated over Antarctica is crucial to decipher, simulate and predict the surface mass balance of the ice sheet (Agosta et al., 2019). Moreover as the sublimation process significantly affects

the isotopic composition of snowfall, this phenomenon should potentially be taken into account and quantified when analyzing the isotopic composition of the near surface water vapor or ice cores, especially in katabatic sectors of Antarctica (Bréant et al., 2019; Goursaud et al., 2019).

The present paper aims to study the synoptic conditions and the atmospheric moisture pathways that are associated to either surface precipitation or virga over DDU station in coastal Adélie Land. More precisely, the manuscript tackles the following

questions:

- Which synoptic circulation features drive the advection of moisture and the precipitation over DDU and what are the differences, if any, between surface precipitation and virga cases?

- What are the atmospheric transport pathways of moisture that precipitates above DDU during either virga or surface precipitation cases?



– What is the origin of the moisture contained in snowfall that precipitates or sublimates over DDU?

To answer these questions, we use a combination of ground-based profiling radar and lidar measurements at DDU, Eulerian diagnostics of extratropical cyclones and fronts, Lagrangian back-trajectories from atmospheric reanalysis as well as a trajectory-based moisture source diagnostic scheme. The paper is organized as follows: Section 2 describes the data sets and the methods. Section 3 depicts the analysis of one particular precipitation case at DDU to illustrate the methodology and the typical structure of precipitation events. Section 4 presents the results of the statistical analysis of surface precipitation and virga events at DDU. Section 5 closes the article with a conclusion and a summarizing conceptual model paving the way for future analyses.

## 2  Data and methods

### 2.1  Dumont d'Urville station

DDU station is located on Petrels Island, 5 km off the coast of Adélie Land, Antarctica (-66.66°N, 140.00°E). This place is known to experience very frequent and fierce katabatic winds (König-Langlo et al., 1998; Vignon et al., 2019b) inducing a strong low-level sublimation of snowfall (Grazioli et al., 2017b; Vignon et al., 2019c). The annual mean wind speed is approximately 20 m s$^{-1}$ (Parish and Walker, 2012) and typical katabatic winds flow from 120 to 140° (Pettré and Périard, 1996). Standard measurements of meteorological variables (among which 10 m wind speed and direction) are collected all year round by the French meteorological service at a one minute temporal resolution.

### 2.2  APRES3 campaign data

During the Antarctic Precipitation, REmote Sensing from Surface and Space (APRES3) campaign from November 2015 to February 2016, a set of instruments dedicated to the characterization of precipitation was deployed at DDU among them a K-band micro-rain radar (MRR). Since November 2015, the radar has been continuously measuring and it is still operational today. More details about the campaign can be found in Grazioli et al. (2017a) and Genthon et al. (2018).

In addition to the MRR, a 532-nm elastic lidar with polarization-sensitive system was deployed at DDU, giving reliable data from February to September 2017. After integration to reduce the noise level, the temporal (resp. vertical) resolution of lidar backscattering coefficient and depolarization ratio estimates was 10 min (resp. 23 m).

The MRR reflectivity at the lowest reliable radar gate (300 m a.g.l.) has been converted into precipitation rate following Grazioli et al. (2017a) and Durán-Alarcón et al. (2019). Precipitation and virga events have been differentiated as in Durán-Alarcón et al. (2019). In summary, radar measurements have been hourly averaged and if there is significant reflectivity (higher than 0 mm$^6$ m$^{-3}$) in the column above DDU but no signal at the lowest reliable radar gate, the hourly reflectivity profile is classified as 'virga'. If significant reflectivity is detected at the lowest radar gate, the profile is classified as 'surface precipitation'. It is worth noting that the MRR does not measure above 3000 m a.g.l. Subsequently, the instrument does not see potential sublimation occurring above this altitude, suggesting that the number of virga hours is probably underestimated. In the





present study, we consider the 23/11/2015-22/11/2017 two-year period, characterized by 1934 virga hours, and 3397 surface precipitation hours.

Instead of distinguishing virga and surface precipitation cases, one could separate the reflectivity profiles between those with strong sublimation - with a strongly negative vertical gradient of reflectivity in the lowest layers - and profiles with weak
sublimation. In this paper, we keep the virga-surface precipitation distinction for two main reasons. First, we thereby stay in line with the study of Durán-Alarcón et al. (2019) that performs a thorough statistical analysis of the vertical profiles of radar reflectivity above DDU during virga and precipitation cases. We can therefore parallel and put in perspective our work with theirs. Second, classifying the radar profiles depending on different sublimation intensities would imply to set up arbitrary thresholds on arbitrary parameters (i.e. the mean vertical gradient of reflectivity over a given altitude range), making our results
less objective and our methodology less replicable for other contexts.

One may also question the added value of using MRR ground-based measurements to quantify precipitation intensity and to identify surface precipitation and virga cases instead of using outputs from a state-of-the art reanalysis product, especially ERA5 that exhibits the best Antarctic accumulation fields according to Gossart et al. (2019). A comparison between MRR data and ERA5 is presented in Appendix A. It particularly reveals a large overestimation of the number of hours with light
precipitation in ERA5 compared to radar measurements, precluding a reliable identification of precipitation occurrences from reanalyses.

## 2.3   Cyclone detection algorithm

To characterize the extratropical cyclones transiting over the Austral Ocean, we applied the algorithm of Wernli and Schwierz (2006) and verified by Sprenger et al. (2017) to the ERA5[1] reanalysis data. The algorithm identifies each cyclone as a local
minimum in the sea level pressure field. It then finds the outermost closed sea level pressure contour for each cyclone to estimate its area. More details about the methodology can be found in Wernli and Schwierz (2006) and Sprenger et al. (2017).

### 2.3.1   Front detection algorithm

To locate and characterize synoptic atmospheric front, a slightly modified version of the front detection algorithm developed by Jenkner et al. (2010) was applied on ERA5 reanalysis data as in Sprenger et al. (2017). In summary, the front detection
algorithm evaluates the frontal lines in a smoothed 700 hPa equivalent potential temperature ($\theta_e$) field. $\theta_e$ is conserved during moist adiabatic motions and thus reflects both the temperature and humidity of an air mass. It is thus a powerful variable for identifying air masses of different origin and with contrasting thermodynamic characteristics (e.g., for separating cold and dry polar air masses from subtropical warm and moist ones). The masking condition used to detect the fronts reads:

$$|\nabla \theta_e| > \kappa \tag{1}$$

$\nabla$ is the gradient operator applied over a 100 km distance and $\kappa$ is the frontal strength minimum threshold set at $4 \ \mathrm{K} \ 100 \ \mathrm{km}^{-1}$. The frontal motion and type are then estimated. More details about the methodology can be found in Sprenger et al. (2017)

---

[1]https://www.ecmwf.int/en/forecasts/datasets/reanalysis-datasets/era5



and Jenkner et al. (2010). Over the Antarctic continent, the frontal structures are chaotic and unreliable because of topography-induced quasi-stationary strong temperature gradients (Schemm et al., 2015). The fronts located in regions where the corresponding topography exceeds 100 m a.s.l. have thus been removed.

### 2.4 Back-trajectories of precipitating air parcels

Air parcel Lagrangian back-trajectories were estimated using LAGRANTO (Wernli and Davies, 1997; Sprenger and Wernli, 2015) applied on the ERA5 reanalysis with a horizontal grid of $0.25^\circ \times 0.25^\circ$ and a hourly temporal resolution. Five-day back-trajectories were calculated for every hour with virga or surface precipitation. Trajectories start (backward in time) at DDU, in a column from the surface level (1000 hPa) to the 300 hPa level, with a 50 hPa vertical step. Temperature, specific humidity, condensed water species (snow, rain, cloud ice, cloud liquid water), as well as the pressure at the boundary layer
top are saved along the trajectories every two hours. As we want to focus on the atmospheric pathway of the water contained in snowfall above DDU, we need to restrict the set of trajectories to those corresponding to the air parcels from which water condenses and precipitates above the station. As a first order approximation, we assume that the so called 'precipitating air parcels' in the ERA5 dataset are those for which the snow water content (SWC) vertical gradient in the column above DDU is significantly negative: $dSWC/dz < SWC_T$ where $SWC_T$ has been set to $-1.5\,10^{-9}$ kg kg$^{-1}$ m$^{-1}$ after inspection of
the distribution of $dSWC/dz$ above DDU (not shown). The underlying reason is that if we neglect the advection - which is a reasonable hypothesis for Antarctic stratiform precipitation as the one over the Antarctic coast and given the $\approx 30$ km horizontal resolution of ERA5 - a negative vertical gradient of SWC typically corresponds to a situation with auto-conversion of ice to snow or snowflake growth through vapor deposition, aggregation or riming (see Vignon et al., 2019a for examples above DDU). Examination of individual vertical profiles of SWC reveals that positive vertical gradients are almost always
located in the katabatic layer, where sublimation occurs. Note that this method is based on the assumption that all vertical SWC profiles have the same shape as the one shown in Fig. 1. A visual inspection of SWC profiles revealed that roughly 10% of these shows different shapes, with few cases having two maxima in the profile. Fig. 1 illustrates significant precipitation generation highlighted in light red (between 800 and 600 hPa) and low level sublimation highlighted in light blue (between 950 and 850 hPa).

### 2.5 Lagrangian moisture source diagnostics

The geographical locations of moisture uptakes along the precipitating air parcels' back-trajectories were then diagnosed using the methodology presented in Aemisegger et al. (2014) and adapted from Sodemann et al. (2008). Along each back-trajectory, the locations of a positive change in the specific humidity with time are identified. Each uptake is then characterized by a weight that corresponds to the respective contribution (percentage) to the specific humidity value at the arrival point of the
trajectory. Note that if the air parcel precipitates along its path, the corresponding quantity in specific humidity is discounted. It is worth mentioning that for some situations, moisture uptakes might be missed due to not long enough back-trajectories (limited to 5 days here). Several precipitating air parcels may contribute to the total moisture precipitating above DDU at each hourly time step. Every hour, a weighting is thus applied to determine the contribution of each precipitating air parcel to the

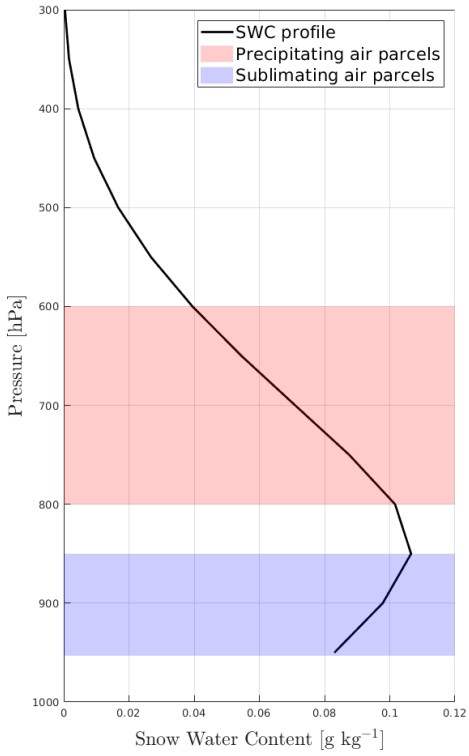

**Figure 1.** Illustration of the precipitating and sublimating layers in a typical hourly SWC vertical profile.

total integrated water vapor of the precipitating column above DDU. Composite maps of moisture uptakes over several time steps can then be produced using the total integrated water vapor of the precipitating column as weighting factor. Note that the algorithm differentiates uptakes occurring within and above the boundary layer. Here, only the moisture uptakes located within the boundary-layer, scaled with a factor 1.5 to account for the uncertainty in the diagnosed boundary layer top from ERA5 (as

5   in Sodemann et al., 2008) were considered.

## 3   Example of a precipitation event at DDU

This section presents a case study of a precipitation event at DDU. This particular event was chosen to illustrate our methodology. It has the advantage of presenting a fully continuous time series of lidar data. The analysis of the event starts on 2000 UTC 7 February 2017 and ends on 2000 UTC 10 February 2017. Fig. 2 shows a meridional cross section of potential temperature,

10   wind and cloud condensates at the DDU longitude and at the beginning of the event (0000 UTC 8 February 2017). The potential temperature field exhibits a typical warm-front structure with the arrival of warm air from the north towards the Antarctic coast associated with a clear ascending stream around 64°S of latitude. The system brings iced clouds and snow (white contours)



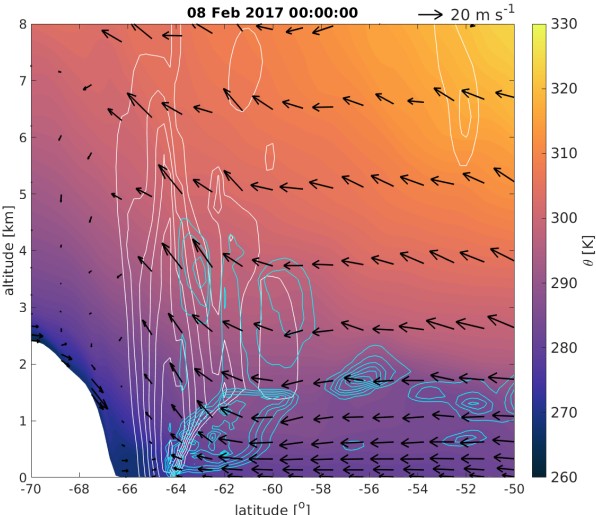

**Figure 2.** 140.00°E meridional cross-section of the potential temperature (shading), meridional and vertical wind (black arrows), snow+ice water contents (white contours), cloud liquid water content (cyan contours) from ERA5 data. Contour interval is $5 \ 10^{-5} \ \mathrm{kg \ kg^{-1}}$. Note that for better readability, the vertical wind component has been multiplied by 100. Remind that DDU is located at 66.66°S.

and supercooled liquid water (cyan contours) while the katabatic flow blows cold air from the Antarctic Plateau towards the station, inducing low-level sublimation.

Three 1h specific times were selected: one during a virga period that precedes precipitation (so-called a 'pre-precipitation virga') at the beginning of the event (0500 UTC 8 February 2017), one during a surface precipitation period (1500 UTC 8 February 2017), and one during a virga period that follows precipitation (so-called a 'post-precipitation virga') at the end of the event (2200 UTC 9 February 2017). Panels a-d of Fig. 3 show how the system evolves above DDU station. The synoptic conditions for the three specific times (identified with vertical red lines) and the precipitating air parcel back-trajectories are plotted in panels e-g. Panels h-j show the time evolution of the pressure and SWC (colors) of the precipitating air parcels.

From 2000 UTC 7 February to 1200 UTC 8 February, the 10-m wind speed at DDU gradually increases (Fig. 3d) in relation with the increase of the ocean-continent pressure gradient owing to the arrival of the extratropical cyclone to the west of DDU. The lidar signal shows the gradual decrease of the cloud height base (Fig. 3a), associated with the warm front arrival (Fig. 3e). At 0500 UTC 8 February, the MRR starts detecting significant reflectivity but the corresponding snowfall does not reach the surface indicating a pre-frontal virga: such features are also observed in mid-latitudes systems (e.g., Clough and Franks, 1991). Precipitating air parcels originate from the north-west and they are advected along the eastern flank of the cyclone (Fig. 3e). In agreement with MRR measurements, all the precipitating air parcels have an arrival pressure level weaker than 700 hPa (Fig. 3h). At 1500 UTC 8 February, there is surface precipitation at DDU (significant MRR signal recorded at 300 m a.g.l.). The synoptic cyclone has moved to the east (Fig. 3f). The warm front has reached DDU, but the cold front remains away from it. The corresponding precipitating air parcel back-trajectories are aligned along a well-defined meridional corridor arriving along



**Figure 3.** Case study of a precipitation event at DDU. Panel a shows a time-height plot of the lidar signal. The 3000 m MRR maximum height is highlighted with a grey line. Mind the lidar signal attenuation during precipitation periods. Panel b is a time-height plot of the MRR reflectivity. Panel c is the time series of the snowfall rate derived from MRR measurements. Panel d shows the time series of the 10-m wind speed and direction. Panels e, f and g show, for three different times indicated with red vertical lines in panels a-d, the sea level pressure (shading), the cyclone mask (white contours), the front lines (red for warm front, cyan for cold front, magenta for indefinite) as well as the 2-day back-trajectories of the precipitating air parcels colored according to their pressure level. The yellow star locates DDU. Panels h, i and j show the corresponding time evolution of the pressure along the back-trajectories. Colors indicate the SWC.




the eastern flank of the cyclone. Fig. 3i shows that most of the precipitating air parcels experience a clear lifting above the ocean associated to a loading in SWC in the last hours before reaching DDU: 8 hours before reaching the station, 8 out of 11 air parcels originate from a height below or close to 900 hPa (cf. Fig. 3i). The higher the parcels arrive above DDU, the earlier they ascend. The lifting occurs perpendicularly to the warm front and just upstream of it. This strongly suggests that the lifting

occurs within - and is due to - the warm conveyor belt of the system. Note that the parcel that arrives above DDU at 300 hPa has been lifted by more than 600 hPa in less than 48 h, fulfilling the criterion of trajectories belonging to a warm conveyor belt by Madonna et al. (2014).

During 9 and 10 February, the cyclone continues its slow eastward transit. One can notice the decrease in wind speed acceleration (Fig. 3d), as the core of the cyclone passes off DDU, and then the decrease of the wind speed owing to the progressive

building-up of the pressure off the station (Naithani et al., 2003). The warm front has passed the station and DDU is in the warm sector where one sees the alternation of hours with relatively weak precipitation affected by sublimation (decrease in reflectivity with decreasing height) and hours with virga. One notices in Fig. 3a the period with surface sublimation from 1100 to 1800 UTC 9 February in between a precipitation period. In a vertical cross-section meridionally crossing the coast at 1200 and 1800 UTC (not shown), a reinforcement of the katabatic flow at DDU is observed. The precipitating air parcels that arrive

at DDU at 2200 UTC 9 February have been trapped by the clockwise cyclonic circulation and they reach the station from the east (Fig. 3g). They arrive closer to the surface and experience a much less pronounced lifting (Fig. 3j). Panel d shows the persistence of strong near-surface wind speed. During 11 February, the extratropical cyclone has moved to the east of the station and its intensity has dramatically decreased.

In the following section, we carry out a statistical analysis over a 2-y period and show that this case study is reasonably well representative of the precipitation events that affect coastal Adélie Land.

## 4   Statistical analysis

### 4.1   Occurrence and timing of virga and surface precipitation at DDU

Over the considered two years of MRR measurements, virga events represented at least 36% of the hourly vertical profiles

with significant reflectivity whereas surface precipitation accounts for 64% (see Sect. A). As the MRR does not measure above 3000 m a.g.l., the actual proportion of virga is most likely even higher. Note that although virga are frequent, the total amount of sublimated snow is higher in cases during which precipitation, albeit affected by sublimation, reaches the ground surface than during virga cases (see Appendix B).

In the previous case study, surface precipitation periods are preceded and followed by virga. One may thus ask the question

whether virga and precipitation cases are always associated to the same synoptic event or not. Fig. 4a and b shows the distribution of the time elapsed between an hour with virga (resp. precipitation) with the closest hour with surface precipitation (resp. virga). The minimum and maximum time delays for the closest virga for precipitation (resp. closest precipitation for virga) are


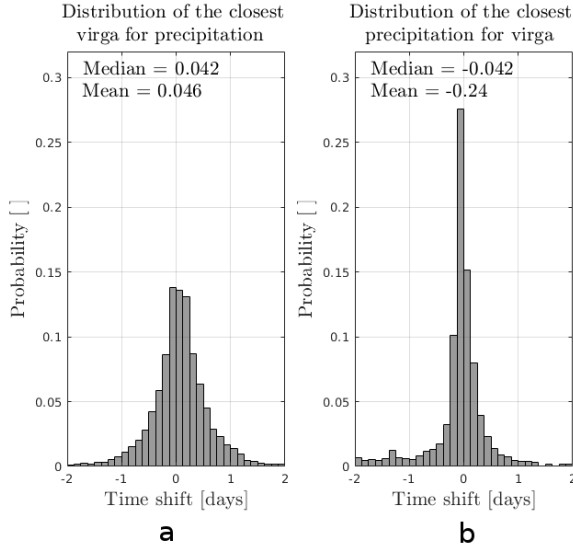

**Figure 4.** Panel a (resp. panel b): two-year distribution of the elapsed time between the closest virga hour (resp. precipitation hour) for each hour with surface precipitation (resp. virga) detected by the MRR. The bin resolution is 3 h. For sake of readability, the axis are truncated to ± 2 days, excluding 0.3 and 13.6 % of the data in panels a and b respectively. In panel a (resp. b), a positive time shift means that the virga (resp. surface precipitation) occurred before the closest precipitation period (resp. virga period).

-3.04 and +2.08 days (resp -9.25 and 12.62 days). A large majority of the virga events occurs close to precipitation periods. Only a few of them can be deemed as isolated event (time difference exceeding ≈ 2 days).

Given that almost all time difference values presented in Fig. 4a and b fall within the [-2, +2] days interval, with a major part between -1 and +1 day, we can reasonably conclude that precipitation and virga hours correspond to different phases of

5 the same synoptic event.

A positive time shift in the closest virga for a considered precipitation (panel a in Fig. 4) means that the virga occurs before the considered precipitation event. Virga precedes and follows precipitation, with more occurrences for pre-precipitation virga, as the distribution in Fig.4a is slightly positively skewed. 60% of virga hours can be classified as 'pre-precipitation' and 40% as 'post-precipitation'.

10 From the case study of Sect.3, one can expect that pre-precipitation virga roughly corresponds to the early phase of pre-cipitation events while post-precipitation virga would be rather associated to the weakening phase of the events. Hereafter, we will investigate the synoptic conditions and the atmospheric moisture transport pathways considering three composites: pre-precipitation virga, surface precipitation and post-precipitation virga.

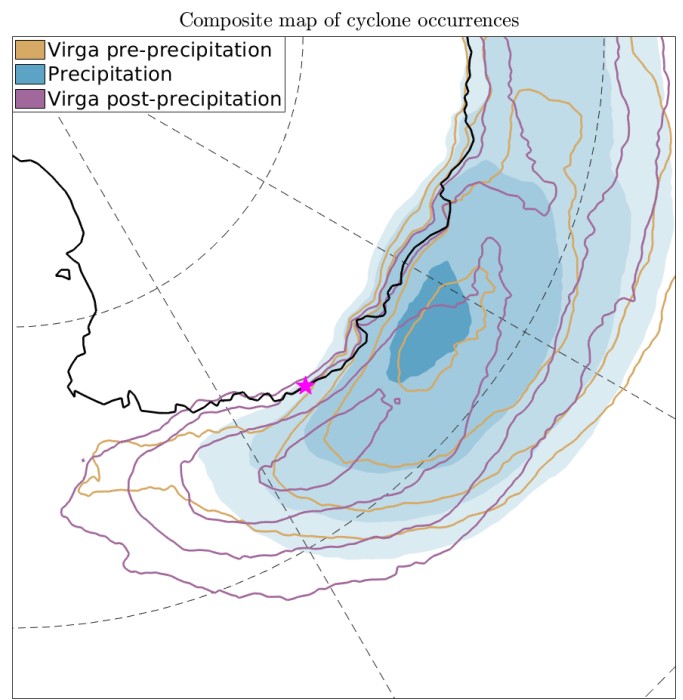

**Figure 5.** Map of the normalized cyclone occurrence from ERA5 data for each of the three composites. An ERA5 grid point is considered to belong to a cyclone if it is included within the cyclone mask identified by the algorithm presented in Sect. 2.3. For sake of readability, only the contours corresponding to the 0.9, 0.95, 0.99, 0.999 quartiles are shown. DDU station is indicated with the purple star.

## 4.2 Synoptic circulation and weather system features

### 4.2.1 Extratropical cyclone position analysis

Precipitation events at DDU are expected to be frequently associated with cyclones, in particular the events with high intensities (Pfahl and Wernli, 2012; Papritz et al., 2014). Fig. 5 shows the statistics of the location of the extratropical cyclones for the

5   three composites.

During precipitation events, an extratropical cyclone is located to the west of DDU (median of the $99.9^{th}$ percentile: $124.5^{o}$ W). The geostrophic flow shows a strong northerly component and the moisture is advected towards Adélie Land along the eastern flank of the cyclone. The position of cyclones for the pre-precipitation virga and surface precipitation composites is reasonably similar (note however a slightly more southerly positioning of the latter). During post-precipitation virga cases, the

10   cyclone has transited towards the east and the geostrophic flow over the station thus becomes easterly. According to Bromwich et al. (2011), such cyclones can be either en route to lysis or they can be reinforced through secondary-development processes.

While the cyclone transit affects the synoptic atmospheric circulation above Adélie Land, 10-m wind measurements at DDU reveal that the regional low-level flow at DDU remains of continental (easterly direction) even during virga or surface





precipitation cases. Therefore, the low-level supply of dry continental air and the potential for precipitation sublimation remains present during all the phases of a precipitation event. It was shown that air is always under-saturated at DDU during surface precipitation events (Vignon et al., 2019c). The wind speed is slightly lower during pre-precipitation virga cases (mean: 9.8 m s$^{-1}$) than during surface precipitation (mean: 13.6 m s$^{-1}$) and post-precipitation virga (mean: 15.2 m s$^{-1}$) cases.

### 4.2.2 Front analysis

In the context of an extratropical cyclone, the associated frontal systems are responsible for a large share of precipitation (Papritz et al., 2014). Large-scale convergence and uplift due to the density difference of the air masses separated by the front induce condensation in the lifted warmer air masses. Panels a and b of Fig. 6 show the maps of occurrence frequencies of warm (panel a) and cold (panel b) fronts identified by the front algorithm on ERA5 data during all precipitation and virga cases at DDU. Panel b in Fig. 6 shows that the cold fronts almost never reach the station during precipitating events. On the other hand, panel a clearly shows that during precipitation and virga at DDU, a warm front is passing over the station. Hence, precipitation and virga at DDU are typically associated with the warm front of a synoptic weather systems that is located to the west of the station.

Panels c, d, and e of Fig. 6 separate the warm front occurrences, allowing us to analyze the evolution of the warm front position during the three phases. The warm front at 700 hPa is generally off DDU during the pre-precipitation virga phase, passes over the station during surface precipitation cases and it has penetrated onto the ice sheet and has moved eastward during post-precipitation virga cases. The above results rely on the ability of the front detection algorithm to properly detect and classify atmospheric fronts. To ensure the robustness of our conclusions, we present in Fig. 7 composite maps of $\theta_e$ at 700 hPa during precipitation and virga periods. Warm air associated with the approaching warm front is located off DDU, while the cold sector of the extratropical cyclones is located to the west of Adélie Land during virga pre-precipitation periods (panel a). During surface precipitation periods (Fig. 7 b), the cold sector has moved northward while the warm sector has traveled eastward and southward over the station, consistently with the expected rotation of the cyclone. This motion is well visible in the difference of equivalent potential temperature between the two composites in Fig. 7d, where the negative blue spot to the north-western side of DDU indicates warmer air in this region due to the warm sector progress in pre-precipitation virga periods compared to surface precipitation periods. During post-precipitation virga (Fig. 7c), the warm sector has penetrated into the Plateau, while relatively cold air is present to the north of DDU suggesting a further rotation of the cold sector. Such a picture overall concurs with our inferences from Fig. 6.

### 4.3 Back-trajectories and lifting of precipitating air parcels

When a synoptic weather system approaches Adélie Land, the occurrence of either pre-precipitation virga, surface precipitation or post-precipitation virga above the coast depends on the location of the extratropical cyclone and in particular of its warm front. We now investigate in more details the atmospheric pathway of the moisture that ultimately precipitates above DDU, with precipitation reaching the ground (precipitation event) or being totally sublimated (virga event).







**Figure 6.** Statistical maps of front occurrences. Panels a and b shows the number of hourly occurrences of warm fronts (panel a) and cold fronts (panel b) during precipitation and virga cases at DDU. Panels c, d and e show the number of hourly occurrences of warm fronts for the pre-precipitation virga composite, surface precipitation composite, and post-precipitation virga composite respectively. Given the different sizes of the three composites, the colorbar has been scaled for each of these three panels. The magenta star locates DDU. Note that fronts located above the Antarctic ice sheet (where the topography exceeds 100 m a.s.l.) have been removed.



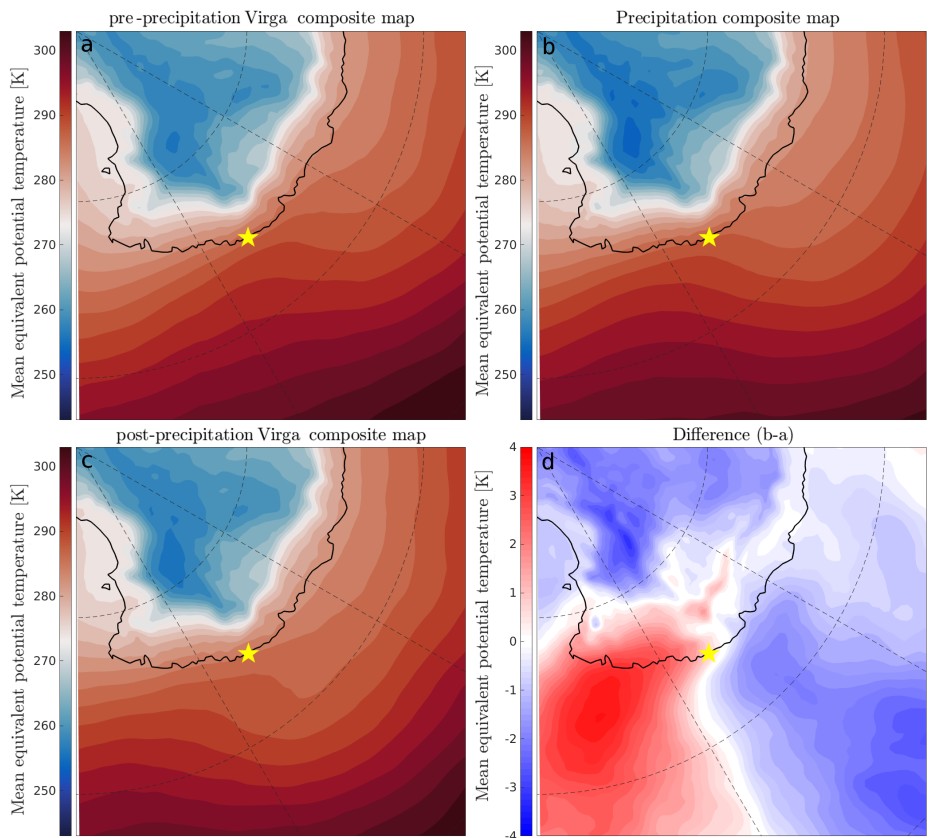

**Figure 7.** Composite maps of equivalent potential temperature at 700 hPa for the pre-precipitation virga composite (panel a), surface precipitation composite (panel b) and post-precipitation virga composite (panel c). Panel d shows the difference between panel b and panel a.

Maps of 2-day back-trajectory occurrences for the three composites are plotted in Fig. 8. During pre-precipitation virga, air parcels follow a clear south-eastward route towards the station while during surface precipitation cases, the general pathway is more meridional. During post-precipitation virga, precipitating air parcels generally reach the station from the north-east. The position of the cyclone in Fig. 5 and the back-trajectories plotted in the case study precipitation in Fig. 3g suggest that air
5 parcels can spin around the cyclone center before being advected over DDU.

The time evolution of the pressure of precipitating air parcels during their transit towards DDU is depicted in Fig. 9. There is no significant difference in the pressure level of the three composites between 48 h and 36 h before reaching DDU (and before). From −36 h, precipitating air parcels experience a pronounced lifting. Air parcels are generally lifted up earlier in time for pre-precipitation and post-precipitation virga compared to surface precipitation cases. Pre-precipitation virga arrive
10 significantly higher above DDU compared to surface precipitation and post-precipitation virga. The distributions of pressure

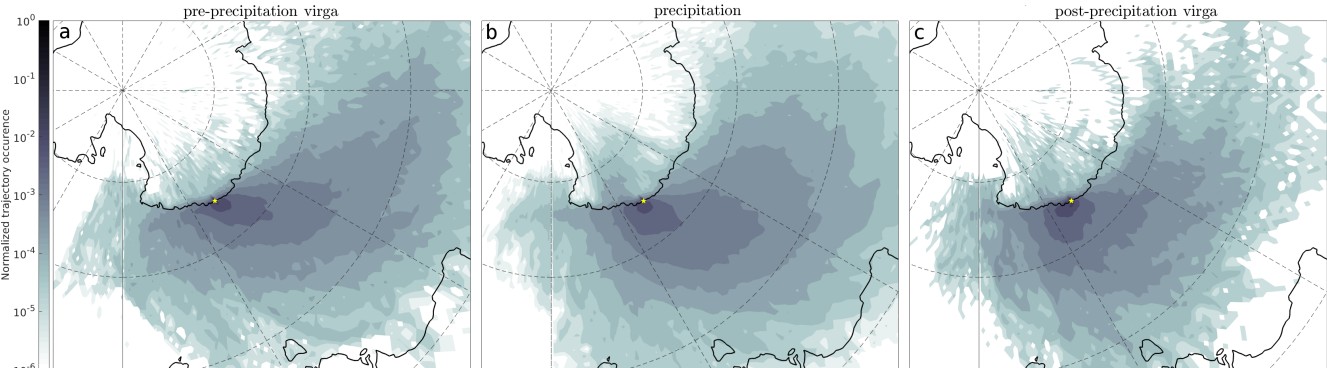

**Figure 8.** Spatial density of the occurrence of 2-day back-trajectories of precipitating air parcels over DDU. Panels a, b and c shows the pre-precipitation virga, surface precipitation and post-precipitation virga composites. Note the logarithmic color scale.

at the arrival above DDU indeed show lower median values (higher altitude) for pre- and post-precipitation virga than for the precipitation composite.

A closer look at the distribution of the change in air parcels' pressure level between different time intervals (not shown) reveals more occurrences of large negative changes (indicating a lifting) in the last six hours before reaching DDU for surface precipitation cases than for pre-precipitation virga and post-precipitation virga cases. The pressure distribution of virga cases is dominated by large negative changes between -6 and -24 h. The pressure distributions of the three ensembles are very similar between 24 and 48 h, and almost identical between 48 and 72 h.

A further analysis of the location of the maximum ascending rate along the back-trajectories (not shown) also reveals closer-to-DDU lifting for the precipitation phase than for virga cases. For the latter, the lifting generally occurs over a broader area and farther away from the station, to the north as well as to the west and east.

These results of the back-trajectory analysis are consistent with the general picture of precipitation induced by air parcel lifting within a warm conveyor belt that crosses a surface warm front moving towards DDU. Precipitating air parcels from pre-precipitation virga events, that generally precedes the front, falls from high air parcels that have been lifted relatively far away from the station when the front was over the Austral Ocean. During surface precipitation cases, the warm front has reached the coast and the snowfall originates from air parcels that were lifted later in time and closer to the continent. Finally, air parcels corresponding to post-precipitation virga generally experience a longer travel time before reaching DDU, explaining their relatively early lifting compared to the surface precipitation composite.

## 4.4 Moisture origin

Different phases of precipitating events over Adélie Land follow significantly different atmospheric pathways during virga and surface precipitation cases. One may also question possible differences in the origin of moisture leading to snowfall, with potential implications for ice core interpretation (Sodemann and Stohl, 2009; Bailey et al., 2019). Composite maps of

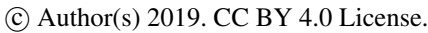



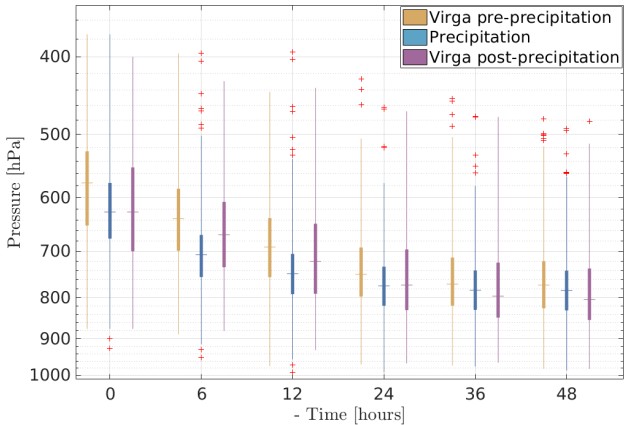

**Figure 9.** Box plot of the three composites of precipitating air parcel pressure for different times along their trajectory towards DDU. On each box, the central mark indicates the median, and the bottom and top edges of the box indicate the 25th and 75th percentiles, respectively.

moisture uptakes along the precipitating air parcels' back-trajectories are presented in Fig. 10. Air parcels corresponding to pre-precipitation virga periods pick up moisture far away from DDU (major uptakes along the $45^{\circ}$S parallel), and to the north-west of the station (close to, and to the east of the $150^{\circ}$E meridian, further south) in the Austral Ocean. These remote moisture sources for pre-precipitation virga cases are collocated with regions known for frequent occurrence of strong large-scale ocean

evaporation events (Aemisegger and Papritz, 2018). On the other hand, air parcels corresponding to actual surface precipitation periods generally pick up moisture closer to DDU at a longitude close to the one of the station. This difference in moisture origin is consistent with the fact that precipitating air parcels for pre-precipitation virga cases are lifted up earlier in time and farther from the station compared to surface precipitation cases (Fig. 9). The large fraction of moisture originating from a region off the coast of DDU during precipitation events points towards a potentially important contribution of rain evaporation

and snow sublimation during precipitation in the warm sector of the cyclone already prior to the air parcel's arrival at DDU. The distinction between surface evaporative sources and recycled moisture by precipitation evaporation in the moisture source diagnostics will be the object of future studies.

For post-precipitation virga back-trajectories, the map of moisture uptakes show a similar pattern to the surface precipitation composite. However, there is no very clear confined regions off DDU that dominates the signal. This might be partly explained

by the smaller size of the post-precipitation virga composite and to its higher sensitivity to particular events compared to the other two.

It is worth noting that the moisture origin of Antarctic precipitation shows a strong seasonality (Sodemann and Stohl, 2009), particularly owing to the presence or absence of sea ice. This effect is illustrated for DDU with maps of seasonal moisture uptakes and sea-ice extent in Appendix C.





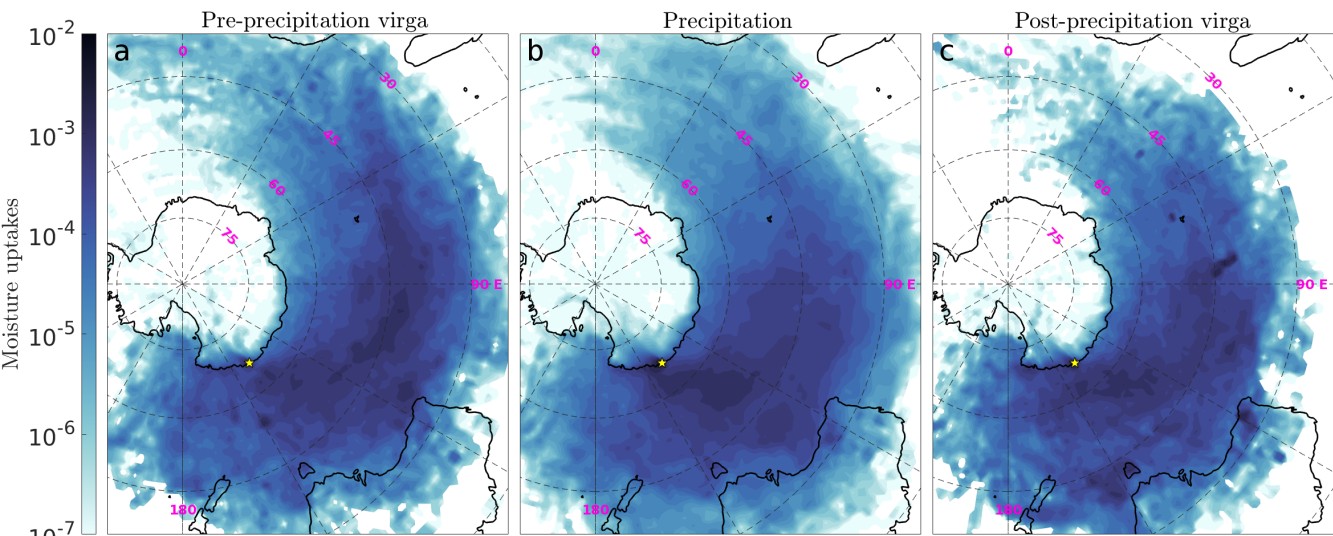

**Figure 10.** Composite maps of moisture uptakes for precipitating air parcels over DDU. The quantity (fractional uptakes km$^{-2}$) shown in the maps is the sum of the contributions to the specific humidity at the arrival of the air parcel over DDU, weighted by the total amount of water vapor contained in precipitating air parcels above the station. Pre-precipitation virga, surface precipitation and post-precipitation cases are shown in panel a, b and c respectively. Note the logarithmic colorscale.

## 5 Discussion and conclusions

This study employs ground-based remotely-sensed measurements, atmospheric back-trajectories, diagnostics of extratropical cyclones, fronts, and moisture sources to characterize the synoptic conditions and the atmospheric moisture transport pathways associated with precipitation and virga cases at DDU, coastal Antarctica. The inspection of vertical profiles of radar reflectivity
made it possible to distinguish hours with actual surface precipitation from hours with virga over a two-year period. It was further shown that surface precipitation and virga actually correspond to different phases of the same synoptic system.

A composite analysis of the pre-precipitation virga, surface precipitation and post-precipitation virga cases leads us to the following conclusions which are pictured in a conceptual model in Fig.11.

- Precipitation and virga at DDU are typically associated with the warm front of an extratropical cyclone that is located to
the west of the station. When the warm front approaches DDU, precipitation falling from high frontal clouds are fully sublimated in the lowest levels of the troposphere due to the remnant low-level katabatic wind. Precipitating air parcels arrive from the south-west and they experience a sharp lifting probably within the warm conveyor belt 24 to 6 hours before reaching the coast. They pick up their moisture over the Austral Ocean in a remote sector to the west of Adélie Land.

- During the surface precipitation phase, the warm front has reached the continent. Precipitation is observed at the surface, as snowfall is intense enough to not completely sublimate even though dry continental winds are still blowing at low-





levels above the coast. Back-trajectories of precipitating air parcels are more meridional and their lifting occurs later in time and closer to the station compared to pre-precipitation virga. They also pick up moisture from the ocean more to the east and closer to the station.

- Post-precipitation virga cases are less frequent than pre-precipitation ones and they often correspond to the weakening phase of snowfall in the warm sector, sometimes in alternation with surface precipitation periods. They occur when the warm front has penetrated into the Plateau, as the cyclone has moved to the east, off the coast of DDU. Precipitating air parcels arrive above DDU from the north east, skirting the eastern flank of the cyclone and experiencing a long travel before reaching the station.

An aspect that was not investigated in this study is the potential seasonality of the precipitation event structure. Over the same analysis period, Durán-Alarcón et al. (2019) reveal a slightly higher virga-to-precipitation ratio in summer compared to the other seasons, probably owing to the warmer temperature and higher saturation vapor pressure implying a higher moisture holding capacity of the air for a given relative humidity. In our study, further seasonal analyses have not revealed major differences in terms of cyclone positions during virga and precipitation cases, although one might expect a signature of the semiannual oscillation with larger and near-coast weather systems in winter than in summer (Uotila et al., 2011). The only clear seasonality signal is the one in the moisture origin (Appendix C) that is directly affected by the sea ice extent. However, a two-year analysis period is not sufficient to draw robust conclusions about a potential seasonality of the structure of precipitation and virga events and this aspect deserves further work on longer MRR time series in the future.

Furthermore, Turner et al. (2019) underline the role of extreme precipitation events in the total surface snow accumulation and snowfall variability in Antarctica, concurring with atmospheric rivers over Dronning Maud Land (Gorodetskaya et al., 2014). At DDU, MRR measurements show that 54% of the accumulated precipitation is explained by the 10% of hours with the highest precipitation rate (higher than the 0.9 quantile). A first analysis showed that 72% of the precipitation events identified as being classified as 'atmospheric river' by the algorithm of Wille et al. (2019) belongs to the 10% (0.9 quantile) most intense hourly radar reflectivity (thus precipitation). In terms of hours, it represent only 27% of the number of hours belonging to this 0.9 quantile, implying that a few hours of precipitation bring large quantity of water for the accumulation over the Antarctic ice sheet. However our analysis period is at the moment too short to draw robust conclusions about intense precipitation events at DDU.

Last but not least, this paper has focused on one particular location of coastal Antarctica. Grazioli et al. (2017b) and Vignon et al. (2019c) showed that low-level sublimation occurs over many sectors along the coast of Antarctica. Moreover, Adélie Land corresponds to a specific location of the southern hemisphere stormtrack, with a major cyclone genesis (resp. lysis) region at the east (resp. west) of DDU (Hoskins and Hodges, 2005). It would be interesting to carry out a similar study for other sectors in Antarctica to assess how representative our conclusions are at the Antarctic scale.





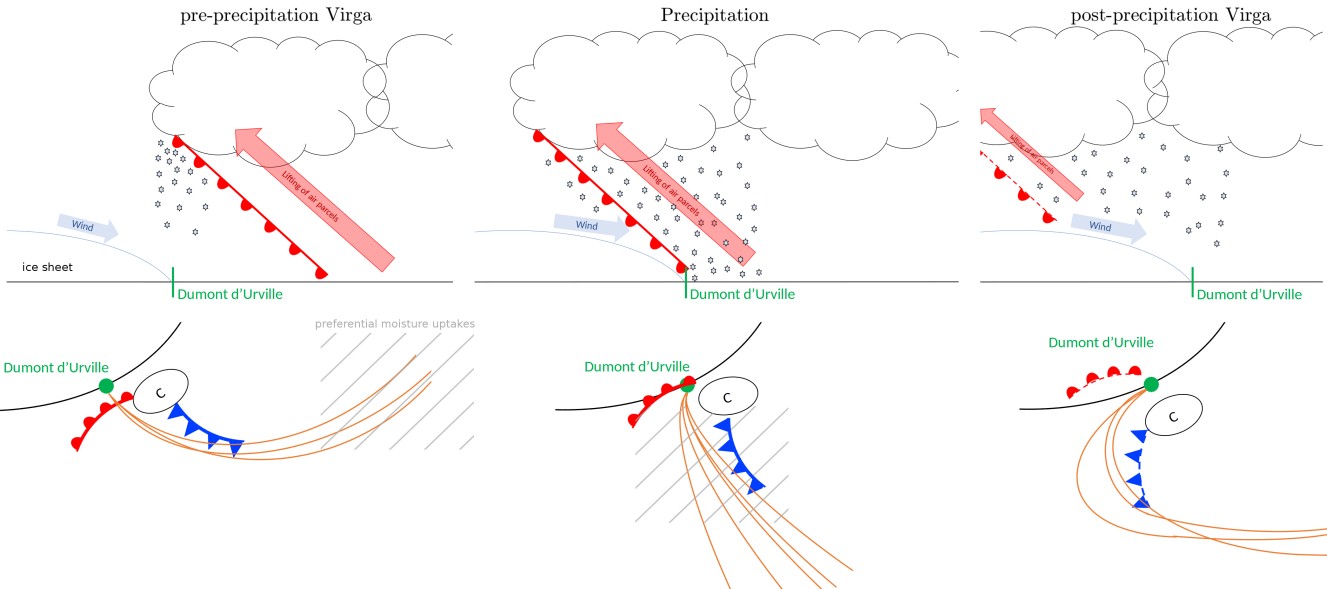

**Figure 11.** Conceptual scheme summarizing the sequence of the different phases of a typical warm front event and associated precipitation over DDU station. The trajectories in orange arrive in the layer where precipitation is formed. In the bottom subplots, south pole is towards the top-left corner direction.

Appendix

## Appendix A: Comparison between the MRR and ERA5 reanalysis

In this part, we briefly compare the surface precipitation in the ERA5 reanalysis at the DDU grid point to the MRR measurements. Note that hourly precipitation rates in ERA5 are always above the lowest hourly precipitation rate measured by the MRR at the first reliable radar gate ($2.10^{-6}$ mm h$^{-1}$). Over the two-year period, the accumulated precipitation in ERA5 is 1383.44 mm, which is close to the accumulated precipitation measured by the MRR: 1335.12 mm.

However, Fig. A1a shows that ERA5 underestimates the relative number of high precipitation intensities. On the other hand, the reanalysis overestimates (note the logarithmic y-axis) the number of low precipitation cases. Moreover ERA5 does not represent hourly precipitation rates exceeding 2.5 mm h$^{-1}$ while the MRR show values up to 4 mm h$^{-1}$.

Fig. A1b further shows that the accumulated surface precipitation is explained by more frequent low-precipitation hours in ERA5 compared to MRR measurements. This result is consistent with the findings of Grazioli et al. (2017a). In addition, 50% of the total surface precipitation measured by the MRR corresponds to the 294 hours (28 days) with the highest precipitation rates while in ERA5, it corresponds to 680 hours (45 days).





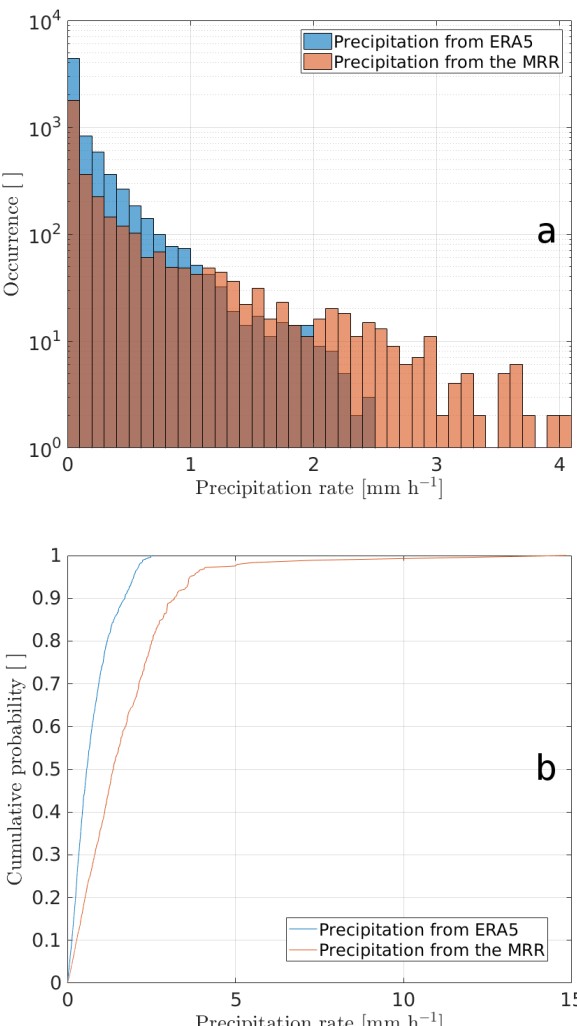

**Figure A1.** Comparison of surface precipitation between MRR and ERA5 over the period 23/11/2015-22/11/2017. Panel a shows the distribution of hourly precipitation rates (note the logarithmic scale in y-axis), panel b the cumulative probability of increasing snowfall rates to the total accumulated precipitation.





|  | E=0 | E=1 | Total | UA |
|---|---|---|---|---|
| M=0 | 10173 | 3960 | 14133 | 71.98% |
| M=1 | 55 | 3342 | 3397 | 98.38% |
| Total | 10228 | 7302 | 17530 | / |
| PA | 99.46% | 45.77% | / | OA = 77.10 % |

**Table A1.** Contingency table comparing hours with precipitation from MRR measurements (M in the table) and from ERA5 (E in the table). Hours with no precipitation are included in the statistics. 1 (resp. 0) means with (resp. no) precipitation event. 'UA' refers to User's Accuracy, 'PA' to Producer's accuracy and 'OA' to Overall Accuracy.

The contingency table (Tab. A1) compares the precipitation occurrences between the MRR and ERA5. More than 98% of the hourly precipitation events measured by the MRR are also present in the ERA5 reanalysis, regardless of the precipitation intensity. On the other hand, only 46% of precipitation hours in ERA5 correspond to actual MRR measurements.

Over the two-year period, there are 10173 hours when both ERA5 and MRR did not detect any precipitation. The overall accuracy is 77.1% between ERA5 and the MRR. This percentage should however be interpreted with caution as it is influenced by the non-precipitation days, which are counted as matching in the overall accuracy calculation.

## Appendix B: Amount of sublimated snow

We provide here a first-order quantification of the amount of sublimated snow using MRR reflectivity data. The sublimated precipitation is calculated by subtracting the maximum snowfall rate in the column with the quantity of precipitation measured at 300 m a.g.l. using the local reflectivity-snowfall rate (Z-S) relationship derived at DDU in Grazioli et al. (2017a). The results are presented in the seventh row of Tab. B1.

The amount of precipitation that sublimates during surface precipitation cases (resp. virga cases) is 791 mm (resp. 125 mm), corresponding to 37% (resp. 100%) of the maximum falling precipitation in the column. Summing surface precipitation and virga periods, 917 mm of precipitation is sublimated, corresponding to 41% of the maximum falling precipitation in the column. This can be compared to the amount of accumulated precipitation at the surface (first reliable radar gate): 1335 mm.

However, such results should be interpreted with care since the Z-S relationship derived for near-surface conditions is used at an altitude up to 3000 m a.g.l.. It may not be valid in the top layers, especially above the aggregation layer, because of different snowflake concentrations and particle types. Yet, the distribution of the maximum reflectively in the column for each event (not shown) reveals that 50% (resp. 95%) of the profiles shows values of maximum reflectivity at an altitude below or equal to 1000 (resp. 1800) m a.g.l.. Even though we cannot provide an uncertainty associated to the vertical extrapolation of the Z-S relationship, we can test different Z-S relations from the literature (Grazioli et al., 2017a). The results are presented in the top six rows of Tab. B1, and the mean and standard deviations of the sublimated snow estimations with the different relationships are





| | Z-S relation | Precipitation at 300m | Sublimated precipitation | | Sublimated virga | | Sublimation precipitation+virga | |
|---|---|---|---|---|---|---|---|---|
| | [mm$^6$ m$^{-3}$]-[mm h$^{-1}$] | [mm] | [mm] | [%] | [mm] | [%] | [mm] | [%] |
| B90A | Z = 67 S$^{1.28}$ | 1501.5 | 804.7 | 34.89 | 244.49 | 100 | 1049.17 | 41.13 |
| B90B | Z = 114 S$^{1.39}$ | 1035.2 | 537.4 | 34.17 | 187.84 | 100 | 725.26 | 41.20 |
| B90C | Z = 136 S$^{1.30}$ | 872.4 | 464.8 | 34.76 | 145.08 | 100 | 609.92 | 41.15 |
| W08A | Z = 28 S$^{1.44}$ | 2797.5 | 1431.4 | 33.85 | 530.36 | 100 | 1961.75 | 41.22 |
| W08B | Z = 36 S$^{1.56}$ | 2309.2 | 1141.1 | 33.07 | 480.81 | 100 | 1621.93 | 41.26 |
| W08C | Z = 48 S$^{1.45}$ | 1923.3 | 981.3 | 33.78 | 367.71 | 100 | 1348.96 | 41.22 |
| **Local DDU** | **Z = 76 S$^{0.91}$** | **1335.1** | **791.4** | **37.22** | **125.46** | **100** | **916.85** | **40.71** |
| Mean ± std | / | / | 879 ± 338 | 34.53 ± 1.34 | 297 ± 164 | 100 ± 0 | 1176 ± 492 | 41.13 ± 0.19 |

**Table B1.** Amount of precipitation and sublimated precipitation using the different Z-S relationships considered in Grazioli et al. (2017a). Percentages correspond to the quantity of precipitation at the 300 m level with respect to the maximum precipitation in the column. The acronym for each Z-S law reference can be found in the Grazioli et al. (2017a).

shown in the bottom row of Tab. B1. Although the magnitude of the amount of total precipitation and sublimated precipitation varies a lot between Z-S relationships, their relative amount is very consitent.

## Appendix C: Seasonal moisture source analysis

Maps of summer (DJF) and winter (JJA) moisture uptakes for surface precipitation cases are plotted in Fig. C1. During sum-
5 mer, when sea ice concentration is minimum, moisture uptakes occur closer to the coast. In winter, when sea ice extends further north, air parcels generally pick up moisture further north in the Austral Ocean, but also less to the east of DDU compared to summer. One can notice the sharp gradient of moisture uptakes at the sea ice margin suggesting the limiting effect of sea ice on evaporation.

*Data availability.* The APRES3 campaign data are freely distributed on the PANGAEA data repository (https://www.pangaea.de, https: //doi.org/10.1594/PANGAEA.883562)

*Author contributions.* NJ, EV and AB designed and conducted the study. MS run the LAGRANTO algorithm and provided the back-trajectories as well as the cyclones and front diagnostics. FA calculated the moisture sources. NJ carried out the major part of the analysis. NJ prepared the manuscript with contributions from all the authors.

*Competing interests.* The authors declare they have no conflict of interest.





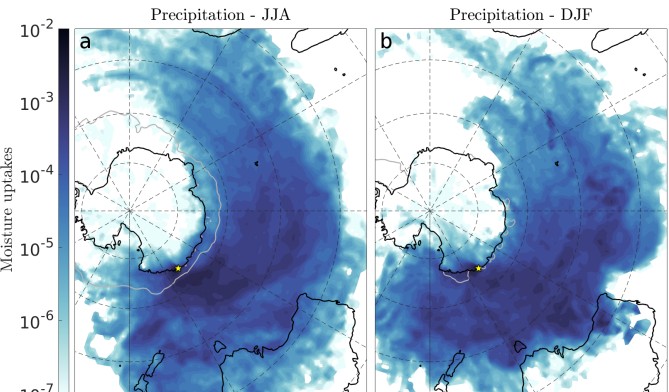

**Figure C1.** Same as Fig. 10a but conditioned to the winter (JJA, panel a) and summer (DJF, panel b) seasons. The grey line shows the $70^{th}$ percentile contour of the seasonal sea ice concentration in ERA5.

*Acknowledgements.* This work has been funded by the EPFL LOSUMEA project. We are grateful to Irina Gorodetskaya for her comments on a preliminary version of this manuscript as well as Iris Thurnherr, Josué Gehring and Alfonso Ferrone for fruitful discussions and support. Météo France is acknowledged for providing the near-surface measurements at DDU station. The authors acknowledge the support of the French National Research Agency (ANR) and of the French Polar Institute to the APRES3 project. Claudio Durán-Alarcón is thanked for
5   pre-processing the radar data. Jonathan Wille and Vincent Favier are gratefully acknowledged for discussions and complementary analyses on atmospheric rivers.



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
