# Peer review of "Synoptic conditions and atmospheric moisture pathways associated with virga and precipitation over coastal Adélie Land in Antarctica"

_The Cryosphere, 2019_

## Referee Comment (RC1) · Anonymous Referee #1 · 14 Feb 2020

The manuscript "Synoptic conditions and atmospheric moisture pathways associated to virga and precipitation over coastal Adélie Land in Antarctica" is a well-motivated study, which clearly presents different synoptic situations leading to virga or precipitation at Dumont d'Urville station (DDU). The paper is scientifically interesting, and provides very useful knowledge of those processes and related atmospheric conditions. The text is nicely written, and mostly easy to follow, although containing quite a lot of detailed information which makes the "main" story more challenging to follow. Anyway, this a paper that deserves to be published, after some minor revision.

General comments:

[Figure]

- I wonder do to pressure systems/fronts affect the strength of katabatic winds at DDU, or are they more or less constant. If the katabatic winds are affected, this might affect occurrence of virga... This could be discussed.

Specific comments:

- Consider moving the content presented in the appendix to Supplementary material, as this information is not fundamental for the main manuscript.

- Page 7, line: instead of "weaker", I recommend to use "lower".

- Page 9, line 2: To me it seems that they are below 900 hPa ∼20 h before, and not 8 h before (if time 0 is the arrival time at DDU, and not 48 h).

- Figure 2: Time of this cross section should be given in the caption. In addition, mark the DDU on the plot (instead of giving the latitude in the caption).

- Figure 2: Instead of "randomly" multiplying the vertical wind component by 100, one could also scale it according to the geometry of the axes. Scaling factor of the horizontal wind vectors would be wind speed/distance of the whole x-axis in km (= 20 latitude degrees in km). Similarly, y-axis scaling factor would be vertical wind/altitude distance shown on the figure (=8 km). If you use these scaling factors to plot your results, they would show the vertical movement more realistically with respect to your potential temperature etc. fields.

- Figure 3: This figure (especially in e, f, and g) contains a lot of information, almost too much. Especially, colors of trajectories (in e, f, g) are not visible enough. In addition, add date on the x-axis of (d) together with the time.

- Figure 6 has varying scale for y-axis due to different composite sizes. It would be clearer to divide the occurrence with the composite size and plot the fraction/percentage of occurrence. This would make the scale to be the same for all the variables and allow for direct comparison of the cases.

---

## Referee Comment (RC2) · Sergi Gonzalez (Referee) · 10 Mar 2020

Sergi Gonzalez (Referee)

sgonzalezh@aemet.es

General Comments This manuscript investigates, using ground-based measurements (MRR) and reanalysis simulations (back-trajectories and diagnostic tools), the synoptic conditions that produce precipitation, both virga and surface snowfall, over Durmont d'Urville station. The authors found that precipitation and virga (pre- and post- precipitation) are associated to different phases of crossing warm fronts. In the study, they identify the synoptic mechanisms that produce such precipitation, showing the importance of large-scale lifting into the warm conveyor belt. Although some sublimation produced by katabatic winds is almost always present in the three phases, what dis-

tinguishes surface precipitation from virga is an enhanced lifting near the station that releases moisture picked up nearby.

Since first MRRs (in Dumont D'Urville and Princess Elisabeth) were recently installed in Antarctica, several advances in the field of Antarctic precipitation have been produced. Now, precipitation and sublimation is been better quantified, and we know better the ability of the reanalyzes to reproduce it. This is another piece of work that improves our knowledge about how moisture is released as precipitation over the Eastern Antarctic coast. In this case, the investigation focuses on the conditions that produces such precipitation. Although most of the results are not unexpected (for example that precipitation is associated with a cyclone west to DDU), they are quantified. In my opinion, the most important finding is showing that the areas where moisture is picked up, is in the Southern Ocean near to the station that may have implications for ice core analysis.

Piecemeal:

The title is clear and clearly define the object of this study. Introduction is well written, and the unifying thread is appropriate. Relevant literature about previous studies about East Antarctic precipitation are cited. The objectives are clear and concise. The several data sources and methods of this paper are clear and correctly described. Results are organized and very well exposed. I like that authors have shown a study case to exemplify the statistical analysis. I think that it enriches the investigation.

Conclusions

I think that the subject of this paper fits the target of The Cryosphere journal. It is furthermore very well written and structured. I think this interesting research should be published in The Cryosphere. I only have very few minor reviews.

Specific Comments

P14 L1 Please, specify in which height start back-trajectories. The statistics is for the lowest back-trajectories or for all the starting heights from 1000 to 300 hPa? If the later,

it would be interesting showing the differences between low-level trajectories, medium-level trajectories and high-level trajectories (maybe in another Appendix), since their pathways may be very different.

Fig 7 Showing Equivalent Potential Temperature at 700 hPa is a good choice to show here because it is high enough to penetrate into the continent but low enough to show the low-level front. However, using the 700 hPa field over the plateau has no much sense since it intercept the terrain showing extrapolated values. The authors should shade (in black for example) the area over 2500 or 3000 m to avoid to distract the reader with the values under the terrain. Figure 7d is a good way to visualize the differences between pre-precipitation virga and precipitation stages, since at first glance, maps look similar despite the differences. I suggest also to show the difference between c-b.

Technical corrections

P10 L3-9 The paragraph "Given the almost..." have only one sentence. Usually paragraph have several sentences and describe one idea. So, I suggest including this paragraph and the following together with the previous one (that starts in the previous page) since the authors are arguing about the same idea.

---

## Author Comment (AC1) · 8 Apr 2020

**Responses to reviews of 'Synoptic conditions and atmospheric moisture pathways associated with virga and precipitation over coastal Adélie Land in Antarctic'**

Nicolas Jullien, Étienne Vignon, Michael Sprenger,
Franziska Aemisegger and Alexis Berne

April 2020

**Review 1**

The manuscript "Synoptic conditions and atmospheric moisture pathways associated to virga and precipitation over coastal Adélie Land in Antarctica" is a well-motivated study, which clearly presents different synoptic situations leading to virga or precipitation at Dumont d'Urville station (DDU). The paper is scientifically interesting, and provides very useful knowledge of those processes and related atmospheric conditions. The text is nicely written, and mostly easy to follow, although containing quite a lot of detailed information which makes the "main" story more challenging to follow. Anyway, this a paper that deserves to be published, after some minor revision.

Dear Referee,
Thank you very much for having carried out a thorough review of our paper and for supporting its publication after revisions. Please find herebelow our responses to your comments.

**General comments**

I wonder do to pressure systems/fronts affect the strength of katabatic winds at DDU, or are they more or less constant. If the katabatic winds are affected, this might affect occurrence of virga... This could be discussed.

   Pressure systems do affect the strength of katabatic winds at DDU (and elsewhere). When they approach the station, the large scale pressure gradient between the continent and the ocean increases. Subsequently, the strength of the katabatic outflow increases as well. Regarding the three considered periods (pre-precipitation virga, surface precipitation and post-precipitation virga),

the mean wind speed is generally higher when the cyclone is off the station and close to it, so during post-precipitation virga cases. However, the wind direction remains easterly or south-easterly during the three phases due to the Coriolis force. It is also worth mentioning that during surface precipitation periods, the near-surface relative humidity remains lower than 100%. Therefore, snowfall is being sublimated over the duration of the three phases of precipitation events. This is illustrated with radiosonde measurements at DDU in the Fig 4 of Vignon et al. 2019b (see Fig. 1 in this document).

To clarify those points in the manuscript, we have renamed the subsection 4.2.1 as 'Extratropical cyclone position and effects on katabatic winds at DDU'. Moreover, we have modified the last paragraph of the subsection as follows:

'The cyclone transit not only affects the synoptic atmospheric circulation above Adélie Land but also strengthens katabatic winds by increasing the continent-to-ocean pressure gradient force. Wind measurements at 10 m at DDU reveal that the wind speed is slightly lower during pre-precipitation virga cases (mean: 9.8 m s−1) than during surface precipitation (mean: 13.6 m s−1) and post-precipitation virga (mean: 15.2 m s−1) cases i.e. when the low-pressure system is closer to the station. However, the regional low-level flow at DDU remains of continental origin (easterly or south-easterly direction) even during virga or surface precipitation cases. In addition, radiosoundings have further revealed that the low troposphere remains under-saturated even during surface precipitation cases (see Fig. 4 of Vignon et al. 2019b). Therefore, the low-level sublimation is effective during over the whole duration of a precipitation event.

**Specific comments**

- Consider moving the content presented in the appendix to Supplementary material, as this information is not fundamental for the main manuscript.

Though not absolutely fundamental, we think that the three sections of the appendix significantly help understand and justify our methodology and give additional - and necessary - information to discuss some of the main results of the paper. Moving the content of those three sections in an external document would make it less visible and could make the paper less easy to follow. That is why we prefer leaving it in the appendix of the main manuscript. We hope the reviewer will concur with our decision.

- Page 7, line: instead of "weaker", I recommend to use "lower".

Pressure and geometrical height vary in the opposite direction vertically. Just using 'lower' might be confusing, as it may suggest a lower geometrical altitude. We have clarified as follows: "In agreement with MRR measurements, all the precipitating air parcels have an arrival pressure lower than 700 hPa, thus higher in altitude (Fig. 3h)."

- Page 9, line 2: To me it seems that they are below 900 hPa ≈20 h before, and not 8 h before (if time 0 is the arrival time at DDU, and not 48 h).

It is indeed a mistake, thank you for pointing it out. We have corrected as

[Figure]

Figure 1: Part of Fig 4 from Vignon et al. 2019b. Vertical profiles of the wind speed (top row) and relative humidity with respect to ice (bottom row) from radiosonde measurements at DDU. Data sets are restricted to precipitation cases. Black lines are the medians, colored lines refer to the 10th, 20th, 30th, 40th, 60th, 70th, 80th and 90th percentiles. In the legend, "Pctx" refers to the shaded area that covers x percent of the data greater than the median and x percent of the data lower than it. The altitude z is above ground level. Wind roses (conditioned to precipitation events) at z = 500 and z = 2000 m are plotted in the lower row panels.

follows: '30 hours before reaching the station, 8 out of 11 air parcels originate from a height below the 900 hPa level (cf. Fig. 3i).'

- Figure 2: Time of this cross section should be given in the caption. In addition, mark the DDU on the plot (instead of giving the latitude in the caption).

Thank you for these suggestions. We have modified the figure accordingly. (Fig.2)

- Figure 2: Instead of "randomly" multiplying the vertical wind component by 100, one could also scale it according to the geometry of the axes. Scaling factor of the horizontal wind vectors would be wind speed/distance of the whole x-axis in km (= 20 latitude degrees in km). Similarly, y-axis scaling factor would be vertical wind/altitude distance shown on the figure (=8 km). If you use these scaling factors to plot your results, they would show the vertical movement more realistically with respect to your potential temperature etc. fields.

We agree with your suggestion and we have changed the figure accordingly (Fig.2). The caption of the figure has been updated as follows: '*140.00° E meridional cross-section of the potential temperature (shading), meridional and vertical wind (black arrows), snow+ice water contents (white contours), cloud liquid water content (cyan contours) from ERA5 data on Feb 08 2017 00:00:00 UTC. Contour interval is $5 \ 10^{-5}$ kg kg$^{-1}$. Horizontal and vertical components of the wind vectors are scaled with the same ratio as the horizontal and vertical extents of the transect. DDU location is indicated by the red circle (140.00° E, 66.66° S).*'.

- Figure 3: This figure (especially in e, f, and g) contains a lot of information, almost too much. Especially, colors of trajectories (in e, f, g) are not visible enough. In addition, add date on the x-axis of (d) together with the time.

We agree with your point. Now trajectories in panels e, f and g are simply plotted in green and we have removed the information on their pressure. Following your recommendation, we have also added the date together with time on the x-axis of panel d. See Fig. 3

- Figure 6 has varying scale for y-axis due to different composite sizes. It would be clearer to divide the occurrence with the composite size and plot the fraction/percentage of occurrence. This would make the scale to be the same for all the variables and allow for direct comparison of the cases.

We agree with your point. We have changed the figure accordingly (cf. Fig. 4).

[Figure]

Figure 2: 140.00ºE meridional cross-section of the potential temperature (shading), meridional and vertical wind (black arrows), snow+ice water contents (white contours), cloud liquid water content (cyan contours) from ERA5 data on Feb 08 2017 00:00:00 UTC. Contour interval is $5 \ 10^{-5}$ kg kg$^{-1}$. Horizontal and vertical components of the wind vectors are scaled with the same ratio as the horizontal and vertical extents of the transect. DDU location is indicated by the red circle (140.00ºE, 66.66ºS).

[Figure]

Figure 3: Case study of a precipitation event at DDU. Panel a shows a time-height plot of the lidar signal. The 3000 m MRR maximum height is marked with a grey line. Mind the lidar signal attenuation during precipitation periods. Panel b is a time-height plot of the MRR reflectivity. Panel c is the time series of the snowfall rate derived from MRR measurements. Panel d shows the time series of the 10-m wind speed and direction. Panels e, f and g show, for three different times indicated with red vertical lines in panels a-d, the sea level pressure (shading), the cyclone mask (white contours), the front lines (red for warm front, cyan for cold front, magenta for undefined) as well as the 2-day back-trajectories of the precipitating air parcels (in green). The yellow star locates DDU. Panels h, i and j show the corresponding time evolution of the pressure along the back-trajectories. Colors indicate the snow water content.

[Figure]

Figure 4: Statistical maps of front occurrences. Panels a and b shows the frequency of hourly occurrences of warm fronts (panel a) and cold fronts (panel b) during precipitation and virga cases at DDU. Panels c, d and e show the frequency of hourly occurrences of warm fronts for the pre-precipitation virga composite, surface precipitation composite, and post-precipitation virga composite respectively. The magenta star locates DDU. Note that fronts located above the Antarctic ice sheet (arbitrarily defined as where the topography exceeds 100 m a.s.l.) have been removed.

---

## Author Comment (AC2) · 8 Apr 2020

**Responses to reviews of**
**'Synoptic conditions and atmospheric moisture pathways associated with virga and precipitation over coastal Adélie Land in Antarctic'**

Nicolas Jullien, Étienne Vignon, Michael Sprenger,
Franziska Aemisegger and Alexis Berne

April 2020

**Review 2**

**General Comments**

This manuscript investigates, using ground-based measurements (MRR) and re-analysis simulations (back-trajectories and diagnostic tools), the synoptic conditions that produce precipitation, both virga and surface snowfall, over Durmont d'Urville station. The authors found that precipitation and virga (pre- and post-precipitation) are associated to different phases of crossing warm fronts. In the study, they identify the synoptic mechanisms that produce such precipitation, showing the importance of large-scale lifting into the warm conveyor belt. Although some sublimation produced by katabatic winds is almost always present in the three phases, what distinguishes surface precipitation from virga is an enhanced lifting near the station that releases moisture picked up nearby. Since first MRRs (in Dumont D'Urville and Princess Elisabeth) were recently installed in Antarctica, several advances in the field of Antarctic precipitation have been produced. Now, precipitation and sublimation is been better quantified, and we know better the ability of the reanalyzes to reproduce it. This is another piece of work that improves our knowledge about how moisture is released as precipitation over the Eastern Antarctic coast. In this case, the investigation focuses on the conditions that produces such precipitation. Although most of the results are not unexpected (for example that precipitation is associated with a cyclone west to DDU), they are quantified. In my opinion, the most important finding is showing that the areas where moisture is picked up, is in the Southern Ocean near to the station that may have implications for ice core analysis.

**Piecemeal**   The title is clear and clearly define the object of this study. Introduction is well written, and the unifying thread is appropriate. Relevant

literature about previous studies about East Antarctic precipitation are cited. The objectives are clear and concise. The several data sources and methods of this paper are clear and correctly described. Results a reorganized and very well exposed. I like that authors have shown a study case to exemplify the statistical analysis. I think that it enriches the investigation.

**Conclusions** I think that the subject of this paper fits the target of The Cryosphere journal. It is furthermore very well written and structured. I think this interesting research should be published in The Cryosphere. I only have very few minor reviews.

We gratefully thank Sergi Gonzalez for the thorough review of our article and for supporting its publication in The Cryosphere journal. Please find herebelow our responses to his comments:

**Specific Comments**

-P14 L1 Please, specify in which height start back-trajectories. The statistics is for the lowest back-trajectories or for all the starting heights from 1000 to 300 hPa? If the later, it would be interesting showing the differences between low-level trajectories, medium-level trajectories and high-level trajectories (maybe in another Appendix), since their pathways may be very different.

The statistics presented here are for precipitating air parcels (see Fig.2 in the paper for the selection of precipitating air parcels). The origin pressure (altitude) of tracked air parcels thus depends on the vertical gradient of the snow water content at DDU, and can be different from one timestep to another during a precipitation event. Note that most precipitating air parcels arrive above DDU in the mid troposphere (pressure generally comprised between 550 and 700 hPa, see Fig. 9 of the paper). Including all the back-trajectories in the statistics would of course modify the maps of back-trajectory occurrences in Fig. 8. In particular, including trajectories arriving in the low-level dry katabatic layer at DDU would lead to an increase of trajectory occurrences over the Plateau because katabatic air parcels originate from the Plateau (not shown). Note however that our point here is not to study the trajectory of air parcels according to their arrival height at DDU but to observe the general pathway and origin of parcels that lead to precipitation generation above DDU during the pre-precipitation virga, surface precipitation and post-precipitation virga periods. To clarify this point, we have modified the introduction of Fig. 8 in the text as follows: 'Maps of 2-day back-trajectory occurrences of precipitating air parcels (see Fig. 2 for their selection) for the three composites are plotted in Fig. 8. It should be noted that trajectories of non-precipitating air parcels - among which those arriving in the low-level dry katabatic layer - are not included in the statistics.'

-Fig 7 Showing Equivalent Potential Temperature at 700 hPa is a good choice to show here because it is high enough to penetrate into the continent but low

[Figure]

Figure 1: Composite maps of equivalent potential temperature at 700 hPa for the pre-precipitation virga composite (panel a), surface precipitation composite (panel b) and post-precipitation virga composite (panel c). Panel d shows the difference between panel b and panel a, panel e the difference between panel c and panel b. Regions with a topography higher than 2500 m ($\approx$ altitude of the 700 hPa level) have been shaded.

enough to show the low-level front. However, using the 700 hPa field over the plateau has no much sense since it intercept the terrain showing extrapolated values. The authors should shade (in black for example) the area over 2500 or 3000 m to avoid to distract the reader with the values under the terrain.

Following your recommendation, we have shaded the regions with an altitude higher than 2500 m.

-Figure 7d is a good way to visualize the differences between pre-precipitation virga and precipitation stages, since at first glance, maps look similar despite the differences. I suggest also to show the difference between c-b. Following your recommendation, we also show the difference between c-b (Panel e in Fig. 1). We have also modified the end of Sect. 4.2.2 as follows: 'During post-precipitation virga (Fig. 7c), the warm sector has penetrated into the Plateau, while relatively cold air is now present to the north and east of DDU suggesting a further rotation of the cold sector of the cyclone (Fig. 7e). Such a picture overall concurs with our inferences from Fig. 6.'

- P10 L3-9 The paragraph 'Given the almost...' have only one sentence. Usually paragraph have several sentences and describe one idea. So, I suggest including this paragraph and the following together with the previous one (that

starts in the previous page) since the authors are arguing about the same idea
This has been done.

---

## Author Comment (AC3) · 8 Apr 2020

**Responses to reviews on**
**'Synoptic conditions and atmospheric moisture pathways associated with virga and precipitation over coastal Adélie Land in Antarctic'**

Nicolas Jullien, Étienne Vignon, Michael Sprenger,
Franziska Aemisegger and Alexis Berne

March 2020

Dear The Cryosphere Editor,
Please find in this document our answers to the referees' comments. We hope that our corrections to the manuscript will make it suitable for publication in The Cryosphere.

**1 Review 1**

The manuscript "Synoptic conditions and atmospheric moisture pathways associated to virga and precipitation over coastal Adélie Land in Antarctica" is a well-motivated study, which clearly presents different synoptic situations leading to virga or precipitation at Dumont d'Urville station (DDU). The paper is scientifically interesting, and provides very useful knowledge of those processes and related atmospheric conditions. The text is nicely written, and mostly easy to follow, although containing quite a lot of detailed information which makes the "main" story more challenging to follow. Anyway, this a paper that deserves to be published, after some minor revision.

Dear Referee,
Thank you very much for having carried out a thorough review of our paper and for supporting its publication after revisions. Please find below our responses to your comments.

**General comments**

I wonder do to pressure systems/fronts affect the strength of katabatic winds at DDU, or are they more or less constant. If the katabatic winds are affected, this might affect occurrence of virga... This could be discussed.

Cyclones and their fronts do affect the strength of katabatic winds at DDU. When they approach the station, the large scale pressure gradient between the continent and the ocean increases (see e.g. Parish and Bromwich, 1998, Papritz et al., 2015). Subsequently, the strength of the katabatic outflow increases as well. Regarding the three considered periods (pre-precipitation virga, surface precipitation and post-precipitation virga), the mean wind speed is generally higher when the cyclone is close to the station just off the coast, so during post-precipitation virga cases. In contrast to the wind speed, the wind direction remains easterly or south-easterly during the three phases. The sustained katabatic outflow during precipitation events is highlighted by lower than 100% near-surface relative humidity observed during surface precipitation periods. Therefore, snowfall is being sublimated over the duration of the three phases of precipitation events. This is illustrated with radiosonde measurements at DDU in the Fig 4 of Vignon et al. 2019b (see Fig. 1 in this document).

To clarify those points in the manuscript, we have renamed the subsection 4.2.1 as 'Extratropical cyclone position and effects on katabatic winds at DDU'. Moreover, we have modified the last paragraph of the subsection as follows:

'The cyclone transit not only affects the synoptic atmospheric circulation above Adélie Land but also strengthens katabatic winds by increasing the ocean-to-continent pressure gradient. 10-m wind measurements at DDU reveal that the wind speed is slightly lower during pre-precipitation virga cases (mean: 9.8 m s$^{-1}$) than during surface precipitation (mean: 13.6 m s$^{-1}$ and post-precipitation virga (mean: 15.2 m s$^{-1}$) cases, i.e. when the low-pressure system is closer to the station. This finding is in agreement with other studies on the link between extratropical cyclones and the formation of cold air outbreaks from katabatic outflows from Antarctica (see e.g. Parish and Bromwich, 1998, Papritz et al., 2015). However, the regional low-level flow at DDU remains of continental origin (easterly or south-easterly direction) during both virga and surface precipitation cases. In addition, radiosoundings have further revealed that the lower troposphere remains under-saturated even during surface precipitation cases (see Fig. 4 of Vignon et al. 2019b). Therefore, the low-level sublimation is effective during the whole precipitation event.'

**Specific comments**

- Consider moving the content presented in the appendix to Supplementary material, as this information is not fundamental for the main manuscript.

Though not absolutely fundamental, we think that the three sections of the appendix significantly help to understand and justify our methodology and give additional - but necessary - information to discuss some of the main results of the paper. Moving the content of those three sections to an external document would make it less visible and could make the paper less easy to follow. That is why we prefer leaving it in the appendix of the main manuscript. We hope the reviewer concurs with our decision.

- Page 7, line 14: instead of "weaker", I recommend to use "lower".

[Figure]

Figure 1: Part of Fig 4 from Vignon et al. 2019b. Vertical profiles of the wind speed (top row) and relative humidity with respect to ice (bottom row) from radiosonde measurements at DDU. Data sets are restricted to precipitation cases. Black lines are the medians, colored lines refer to the 10th, 20th, 30th, 40th, 60th, 70th, 80th and 90th percentiles. In the legend, "Pctx" refers to the shaded area that covers x percent of the data greater than the median and x percent of the data lower than it. The altitude z is above ground level. Wind roses (conditioned to precipitation events) at z = 500 and z = 2000 m are plotted in the lower row panels.

Pressure and geometrical height vertically vary in the opposite direction. Just using 'lower' might be confusing, as it may suggest a lower geometrical altitude. We have clarified as follows: "In agreement with MRR measurements, all the precipitating air parcels have an arrival pressure lower than 700 hPa, thus higher than $\approx 2500$m a.s.l. (Fig. 3h)."

- Page 9, line 2: To me it seems that they are below 900 hPa $\approx 20$ h before, and not 8 h before (if time 0 is the arrival time at DDU, and not 48 h).

It is indeed a mistake, thank you for pointing it out. We have corrected the text as follows: '30 hours before reaching the station, 8 out of 11 air parcels originate from a height below the 900 hPa level (cf. Fig. 3i).'

- Figure 2: Time of this cross section should be given in the caption. In addition, mark the DDU on the plot (instead of giving the latitude in the caption).

Thank you for these suggestions. We have modified the figure accordingly. (Fig.2)

- Figure 2: Instead of "randomly" multiplying the vertical wind component by 100, one could also scale it according to the geometry of the axes. Scaling factor of the horizontal wind vectors would be wind speed/distance of the whole x-axis in km (= 20 latitude degrees in km). Similarly, y-axis scaling factor would be vertical wind/altitude distance shown on the figure (=8 km). If you use these scaling factors to plot your results, they would show the vertical movement more realistically with respect to your potential temperature etc. fields.

We agree with your suggestion and we have changed the figure accordingly (Fig.2). The caption of the figure has been updated as follows: '$140.00°E$ meridional cross-section of the potential temperature (shading), meridional and vertical wind (black arrows), snow+ice water contents (white contours), cloud liquid water content (cyan contours) from ERA5 data on Feb 08 2017 00:00:00 UTC. Contour interval is $5\ 10^{-5}$ kg kg$^{-1}$. Wind vectors are scaled according to the spatial extent of the caption. DDU location is indicated by the red circle $(140.00°E, 66.66°S)$.'.

- Figure 3: This figure (especially in e, f, and g) contains a lot of information, almost too much. Especially, colors of trajectories (in e, f, g) are not visible enough. In addition, add date on the x-axis of (d) together with the time.

We agree with your point. Now trajectories in panels e, f and g are simply plotted in green and we have removed the information on their pressure. Following your recommendation, we have also added the date together with time on the x-axis of panel d. See Fig. 3

- Figure 6 has varying scale for y-axis due to different composite sizes. It would be clearer to divide the occurrence with the composite size and plot the fraction/percentage of occurrence. This would make the scale to be the same for all the variables and allow for direct comparison of the cases.

[Figure]

Figure 2: 140.00ºE meridional cross-section of the potential temperature (shading), meridional and vertical wind (black arrows), snow+ice water contents (white contours), cloud liquid water content (cyan contours) from ERA5 data on Feb 08 2017 00:00:00 UTC. Contour interval is 5 $10^{-5}$ kg kg$^{-1}$. Wind vectors are scaled according to the spatial extent of the figure. DDU location is indicated by the red circle (140.00ºE, 66.66ºS).

[Figure]

Figure 3: Case study of a precipitation event at DDU. Panel a shows a time-height plot of the lidar signal. The 3000 m MRR maximum height is highlighted with a grey line. Mind the lidar signal attenuation during precipitation periods. Panel b is a time-height plot of the MRR reflectivity. Panel c is the time series of the snowfall rate derived from MRR measurements. Panel d shows the time series of the 10-m wind speed and direction. Panels e, f and g show, for three different times indicated with red vertical lines in panels a-d, the sea level pressure (shading), the cyclone mask (white contours), the front lines (red for warm front, cyan for cold front, magenta for indefinite) as well as the 2-day back-trajectories of the precipitating air parcels (in green). The yellow star locates DDU. Panels h, i and j show the corresponding time evolution of the pressure along the back-trajectories. Colors indicate the SWC.

We agree with your point. We have changed the figure accordingly (cf. Fig. 4).

[Figure]

Figure 4: Statistical maps of front occurrences. Panels a and b shows the frequency of hourly occurrences of warm fronts (panel a) and cold fronts (panel b) during precipitation and virga cases at DDU. Panels c, d and e show the frequency of hourly occurrences of warm fronts for the pre-precipitation virga composite, surface precipitation composite, and post-precipitation virga composite respectively. The magenta star locates DDU. Note that fronts located above the Antarctic ice sheet (where the topography exceeds 100 m a.s.l.) have been removed